# Recreational Use of Nitrous Oxide as a Source of Frostbite Injuries to the Skin: A Review of the Literature and a Case Report

**DOI:** 10.3390/ebj6010014

**Published:** 2025-03-07

**Authors:** Sebastian Holm, Reza Tabrisi, Johann Zdolsek

**Affiliations:** Department of Plastic and Reconstructive Surgery, Örebro University Hospital, Faculty of Medicine and Health, Örebro University, SE 70182 Örebro, Sweden; reza.tabrisi@oru.se (R.T.); johann.zdolsek@gmail.com (J.Z.)

**Keywords:** frostbites, frostbite injuries, nitrous oxide, laughing gas

## Abstract

Nitrous oxide has a wide range of medical applications, such as being used as an analgesic in general anesthesia, dental procedures, childbirth and sedation. Lately, it has also been employed as an inhalant recreational drug to induce brief euphoria. Recent studies indicate a worldwide rise in the incidence of skin frostbites associated with nitrous oxide use. A scoping review was conducted to synthesize and summarize the existing literature published in English regarding frostbite injuries associated with the recreational use of nitrous oxide. The literature search was carried out in July 2024 using databases such as Embase, Web of Science and PubMed^®^. From an initial pool of 83 publications, 8 studies were ultimately selected for full-text review as they met our inclusion criteria for analysis. Additionally, we provide a representative clinical case involving a 21-year-old male who experienced frostbite following skin exposure to nitrous oxide. Most publications on nitrous oxide induced frostbites are from recent years, primarily between 2022 and 2024, with the first case documented in 1996. These injuries are mostly observed in young adults, with a female dominance, and are typically localized to the inner thighs. According to the existing literature, the predominant treatment approach is conservative management, with excision and split-thickness skin grafting (STSG) in the second place. This study represents the first literature review summarizing frostbite injuries to the skin from nitrous oxide misuse. There is a need for enhanced preventive measures to raise public awareness and reduce the incidence of frostbite injuries associated with the recreational use of nitrous oxide.

## 1. Introduction

Hypothermic injuries to the skin, or frostbite, seldom present at burn centres [1,2]. Frostbite is, however, not entirely uncommon in military personnel and mountain climbers exposed to cold or arctic weather conditions. Other risk factors for frostbite include homelessness, psychiatric conditions, high alcohol consumption, drug usage, car accidents or other occupational hazards in cold environments [2,3,4].

Common frostbite injuries usually occur due to cold weather conditions and cold exposure for longer periods of time. As the body re-directs its blood flow from the cold-exposed periphery to centrally placed organs, frostbite injuries mostly occur in the fingers, toes, ears and nose. Normal dermal blood flow is around 250 ML/min, but with hypothermia it can drop to 20–50 ML/min, and at temperatures below 0 °C blood flow stops. A cessation of dermal blood flow then causes progressive and deepening ischemia with cellular damage. This damage is either indirect due to dehydration [5,6,7] or due to freezing with intra- and extracellular crystal formation causing necrosis and cell death [6].

Vasodilatation with inflammatory processes occurs as a defence mechanism in order to ensure heat transport through the cold tissue, which, upon prolonged exposure, eventually also freezes, which increases the area of frostbite necrosis even further [7,8]

Nitrous oxide has many medical applications and has for more than a century found use as an analgesic in general anesthesia for dental procedures, childbirth and sedation [9]. Nitrous oxide is also commonly used at parties to fill balloons. It has, however, been used as an inhalant party drug for the induction of a short euphoria in recent decades [10,11].

Nitrous oxide is generally stored in gas cylinders in either liquid or compressed form under high pressure. The temperature of the liquified gas under normal pressure can be between −55 °C and −88.5 °C. When the gas is released, its temperature rapidly increases to −40 °C [12]. Direct skin contact with cold nitrous oxide directly from the container can cause frostbite depending on the length of exposure as well as exposure to the metal outer casing of the aerosol can by the cooled gas. The time needed for nitrous oxide to cause a deeper frostbite is not known [12].

Recent data suggest a small but globally increased number of skin frostbite cases caused by nitrous oxide [12,13].

## 2. Objective

This scoping review seeks to synthesize and draw conclusions from the existing English-language literature regarding frostbite injuries associated with the recreational use of nitrous oxide. Additionally, we provide a representative clinical case involving a 21-year-old male who experienced frostbite following skin exposure to nitrous oxide.

## 3. Method

### 3.1. Search Strategy

A literature search was performed and structured by a health science librarian. The databases utilized included Embase, Web of Science and PubMed^®^ as of 4 July 2024, with no restrictions on the time frame. The search terms incorporated various synonyms for “frostbites”, “cold injuries”, “cold burn”, “nitrous oxide”, “laughing gas”, and “nitrogen protoxide”. The overall count of scientific publications identified was 383. Following the elimination of duplicates, 182 articles were available for abstract screening. (Table 1). The PICO parameters used were P—recreational use of nitrous oxide; I—exposure to nitrous oxide; C—no/limited nitrous oxide exposure; and O—outcomes of cutaneous frostbite caused by recreational use of nitrous oxide.

### 3.2. Eligibility Criteria

The search was conducted without any language limitations, time constraints or restrictions on study design. The studies included focused on clinical cases involving patients who experienced frostbite to the skin as a result of recreational nitrous oxide use. Studies pertaining to the medical application of nitrous oxide were excluded, as were those involving frostbite injuries occurring in areas other than the skin and the medical use of nitrous oxide.

### 3.3. Study Selection

The abstract screening procedure was conducted utilizing Rayyan software 14. On this platform, a blinded review of the abstracts was executed. Two independent reviewers (S.H. and R.T.) assessed the studies for eligibility without prior knowledge of each other’s evaluations. The initial evaluation focused on the titles and abstracts, which was succeeded by a thorough examination of the full texts to determine the potential eligibility of the selected studies. Any discrepancies between the two reviewers were addressed through discussion until a consensus was achieved. If the two investigators disagreed and a consensus was not reached, a third investigator (J.Z.) was involved to cast the deciding vote.

### 3.4. Data Extraction

The data extraction was conducted using Excel (Microsoft Corp., Redmond, WA, USA). This pre-established spreadsheet encompassed details including the first author, country of publication, study design type, publication year, number of patients involved, average age, gender, frostbite location, treatment administered and outcomes achieved (see Table 2).

## 4. Results

Out of a total of 83 publications, 8 studies were selected for a full-text review as they meeting our inclusion criteria for analysis (see Figure 1). Our literature review included three case reports [15,19,20], four case series [10,12,16,17] and one Letter to the Editor [18]. The countries of publications were the UK (*n* = 3), the Netherlands (*n* = 2), Ireland (*n* = 2) and the USA (*n* = 1). The publication years spanned from 2020 to 2024, with one earlier study dating back to 1996 [19]. The majority of the studies were published between 2023 and 2024. Around 11 patients were injured on average each year between 2020 and 2024. The total number of patients included in the review was 54. The mean patient age was 25 years of age. The studies reported a higher number of female patients (*n* = 23) compared to male patients (*n* = 11). The most frequently affected area for frostbite was the inner thighs (*n* = 23), followed by the upper extremities, including the hands (*n* = 11), and the face (*n* = 3). Conservative wound management was the predominant treatment approach (*n* = 21), although debridement of necrotic tissue and split-thickness skin grafting were employed for deeper frostbite cases (*n* = 11). Most wounds demonstrated healing within a timeframe of 1 to 4 months (see Table 3).

### Case Report

A previously healthy 21-year-old male was admitted to our outpatient clinic due to a frostbite injury to his right shoulder. No exact mechanism or length of exposure of the injury could be obtained due to him being unconscious after the recreational use of nitrous oxide. Through discussions and reasoning with the patient, it was assumed that, when unconscious, he had probably been lying on the cannister, leading to a prolonged exposure to both the released nitrous oxide and the cooled outside of the metal aerosol spray-can. The largest part of the frostbite on his right shoulder measured around 8 × 8 cm with two adjacent satellite lesions measuring 1 × 1 cm; Vaseline–gauze was applied. A week after the initial exposure to nitrous oxide, the wound was necrotic with clear signs of full-thickness skin damage (Figure 2). Wound debridement was performed and Vaseline–gauze applied. One week after debridement, the wound was covered with a split-thickness skin graft (STSG). The graft achieved a 100% take, and the wound was completely healed within one month of the initial injury (Figure 3).

## 5. Discussion

The occurrence of frostbite on the skin following recreational use of nitrous oxide appears to be on the rise, as indicated by various studies [12,15,17]. This pattern has been observed in countries such as the UK, Ireland, the Netherlands and, recently, Sweden as well [20]. In our department, we have noted a significant increase in frostbite cases associated with the recreational use of nitrous oxide. The presented case is one example of a patient have seen in our outpatient clinic after recreational misuse of nitrous oxide. Our literature review reveals that most relevant publications have emerged in recent years, particularly between 2022 and 2024. Notably, the first documented case reported in 1996 by Hwang et al. [18] involved a 54 year old male “sniffing” nitrous oxide. This case is distinct from more recent reports, which predominantly feature a younger demography using nitrous oxide as a party drug. A comparison of the seven additional studies included in our review indicates that such frostbite injuries are most frequently observed in young adults, with a higher prevalence among females, and are typically localized to the inner thighs. The prevailing treatment approach, as supported by the current literature, involves initial conservative management, and excision and split-thickness skin grafting (STSG) in the second place.

Baran et al. [17] reported on treating 20 patients suffering from severe frostbite due to the misuse of nitrous oxide. This is the largest study of describing this specific patient group, although it is lacking detailed information regarding factors such as age, gender, the location of frostbite and the treatment administered to each individual. Notably, it was found that the most frequently affected area was the inner thighs, which aligns with the existing literature. The second largest case series by Chen et al. [12] includes 16 patients, with a mean age of 22.5 years. This series provides more comprehensive data on each patient and their subsequent treatment. The findings of this case series are consistent with the overall conclusions drawn from this literature review regarding gender and the site of injury, indicating that the typical patient is a young female in her twenties sustaining frostbite injuries to the inner thighs.

The inner thighs are the most common site of frostbite injury from nitrous oxide. This is most probably due to prolonged exposure to the metal canister held between the legs while filling balloons with gas [12,15,16,17]. Frostbite injuries following nitrous oxide exposure are frequently deep, with a full-thickness injury often requiring debridement and STSG, compared to frostbite caused by environmental conditions, where the majority of cases are treated conservatively with wound dressings.

## 6. Limitation

In our study, only one patient agreed to participate, and a case series was unfortunately not possible. During the past two years, eight patients have been treated at our unit due to a frostbite injury to the skin due to recreational use of nitrous oxide. Previous studies have described difficulties in follow-up with this group of patients after injuries due to the recreational use of nitrous oxide [12,16,17,19]. These patients often present within a few days, and they often miss or opt-out of planned appointments and are lost to follow-up, compared to other injuries such as burns. The delay in presentation and decreased compliance in follow-up can lead to increased risk of infection and scarring with poor esthetic results [16]. It is notable that we did not find any articles from, e.g., Asia, in our literature review, although this does not preclude the possibility that this is potentially a worldwide issue.

## 7. Conclusions

This study represents the first literature review summarizing frostbite injuries to the skin resulting from nitrous oxide misuse. There is a need for enhanced preventive measures, such as media attention, which may raise public awareness and possibly decrease the incidence of frostbite injuries associated with the recreational misuse of nitrous oxide.

## Figures and Tables

**Figure 1 ebj-06-00014-f001:**
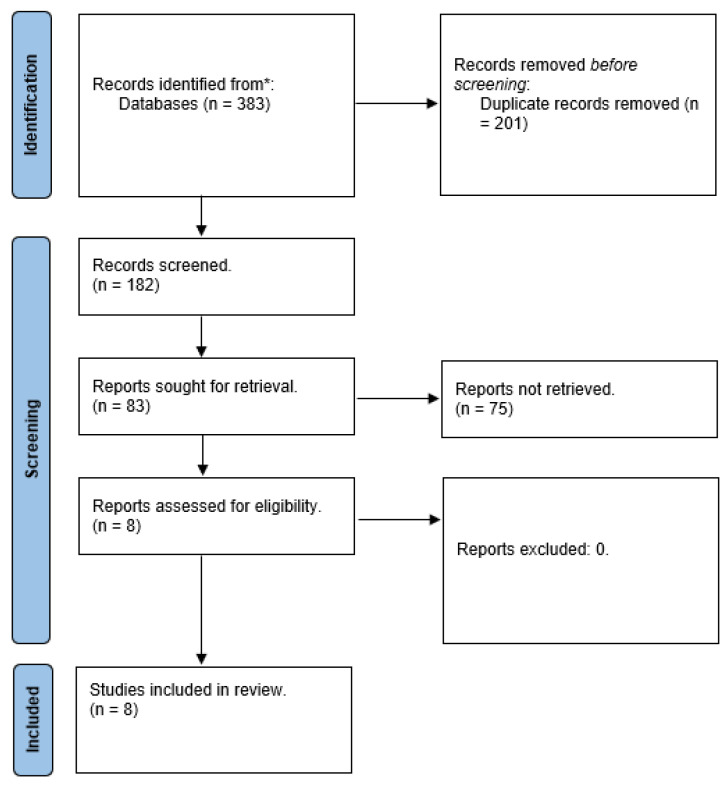
Flow diagram for study selection. *: The databases: Embase, Web of Science, and PubMed^®^.

**Figure 2 ebj-06-00014-f002:**
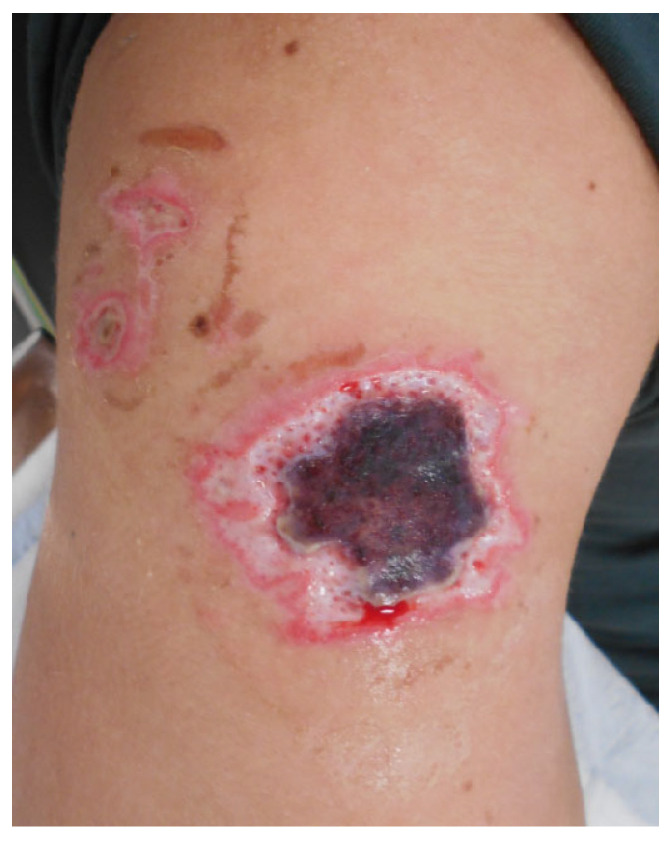
Necrotic tissue, before debridement.

**Figure 3 ebj-06-00014-f003:**
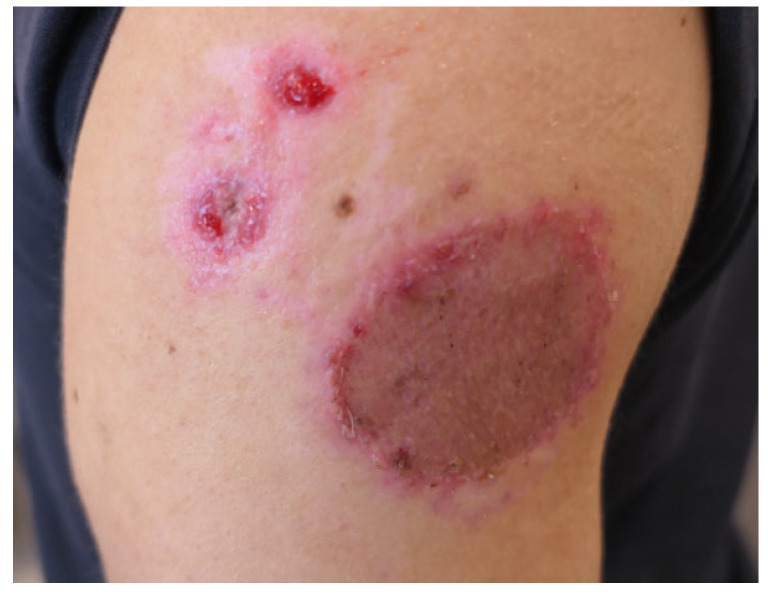
One month after the initial exposure to nitrous oxide and a fully healed STSG.

**Table 1 ebj-06-00014-t001:** The search strategy summarized.

Strategy	Search
Date	4 July 2024
Databases	Pubmed, Web of science and Embase
Search	“Frostbites”, “cold injuries”, “cold burn”, “nitrous oxide”, “laughing gas”, “nitrogen protoxide”
Time limitations	No time limitations; studies until 4 July 2024
Criteria (inclusion/exclusion)	Studies on frostbite injuries to the skin after exposure to nitrous oxide from recreational use All types of study designs Only studies published in English

**Table 2 ebj-06-00014-t002:** Data extraction of included studies based on our inclusion criteria.

First Author	Country	Type of Study	Year of Publication	Nr of Pts Included	Age (Mean)	Gender	Site of Frostbite	Treatment	Outcome
Quax et al. [10]	The Netherlands	CS	2022	2	1: 22 y 2: 18 y	1: M 2: F	1: Inner thigh 2: Inner thigh	Wound debridement, after granulation tissue, split-skin graft transplantation (STSG).	Split-thickness skin graft had an 80% take rate
Allen et al. [14]	Ireland	CR	2024	1	20	M	Forearm	Conservative	Almost fully healed after 4 months
Murphy et al. [15]	Ireland	CS	2024	7	19	1 M 6 F	Inner thigh: 2 Upper extremity: 4 Face: 1	STSG: 2 Conservative: 5	None of the patients had systemic symptoms
Hever et al. [16]	UK	CS	2023	6	23	6 F	Inner thigh: 6	STSG: 3 Conservative: 1 Failed follow-up: 2	All injuries healed within 4 weeks
Chen et al. [12]	UK	CS	2023	16	22.5	7 M 9 F	Inner thigh: 11 Hands: 4 Lip: 1	STSG: 4 Conservative: 12	Healing after surgery: 42.5 days Healing after conservative management: 64 days
Baran et al. [17]	The Netherlands	LE	2020	20	Unknown	Unknown	Majority inner thigh	Unknown	Unknown
Hwang et al. [18]	USA	CR	1996	1	54	M	Left cheek	Conservative	Refused STSG, developed hypertrophic scar
Stone et al. [19]	UK	CR	2021	1	Unknown	F	Inner thigh	Conservative	Healed within 82 days

Nr: number; pts: patients; CS: case series; y: year; M: male; F: female; CR: case report; LE: Letter to the Editor; UK: United Kingdom.

**Table 3 ebj-06-00014-t003:** Summarized data from the included studies.

Data	Results
Study design	Case reports: 3 Case series: 4 Letter to the Editor: 1
Country of publication	UK: 3 Netherlands: 2 Ireland: 2 USA: 1
Year of publication	1996: 1 2020: 1 2021: 1 2022: 1 2023: 2 2024: 2
Total number of patients included	54
Mean age	25 years old
Gender	Females: 23 Males: 11
Site of frostbite	Inner thigh: 23 Upper extremity: 11 Face: 3
Treatment	Conservative: 21 SSGT: 21

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
