# Peer review of "Recreational Use of Nitrous Oxide as a Source of Frostbite Injuries to the Skin: A Review of the Literature and a Case Report"

_2673-1991, 2025, doi:10.3390/ebj6010014_

Round 1
Reviewer 1 Report
Comments and Suggestions for Authors
Thank you for the opportunity to review this manuscript.
1) Line 27 - recommend using "Frostbite is..." as the more common phrasing and consistent with singular use in the remainder of the paper.
2) Line 47 - "leads" should be "lead" and it seems there is a period and new sentence where a comma is intended, recommend revising this.
3) For the eligibility criteria, you state that the search was conducted without and language limitation, however, in the objectives and Table 1 the search is restricted to English. Please clarify
4) Line 113 - this sentence needs to be clarified
5) What was the temporizing measure / wound care that was utilized in the week between the excision of the wound and STSG?
6) In limitations, it is mentioned that there was only one patient that consented to participate. While certainly this would make a case series not feasible, is it not possible to present the total number of such injuries that your center has treated for better context? This data is also necessary to support your assertion in Line 135.
7) In line 46, recommend revising the phrase "on second place"
8) In the discussion, the two largest case series are discussed and you state that these studies are consistent with "the conclusions drawn from this literature review". This stands to reason as they constitute the bulk of the reported patients. recommend removing or revising this section.
9) In comparing frostbite from nitrous oxide to "standard" frostbite, you site a case series from Texas which, in and of itself, represented an unusual series of frostbite in an center unused to seeing it. Further, within that case series, 7/13 patients underwent surgery. This does not support the statement in your paper, please revise.
10) Line 163 - is there data to support this claim?
Author Response
Reviewer 1: Authors comments Manuscript revisions |
|||
Thank you for the opportunity to review this manuscript. 1) Line 27 - recommend using "Frostbite is..." as the more common phrasing and consistent with singular use in the remainder of the paper. 2) Line 47 - "leads" should be "lead" and it seems there is a period and new sentence where a comma is intended, recommend revising this. 3) For the eligibility criteria, you state that the search was conducted without and language limitation, however, in the objectives and Table 1 the search is restricted to English. Please clarify 4) Line 113 - this sentence needs to be clarified 5) What was the temporizing measure / wound care that was utilized in the week between the excision of the wound and STSG? 6) In limitations, it is mentioned that there was only one patient that consented to participate. While certainly this would make a case series not feasible, is it not possible to present the total number of such injuries that your center has treated for better context? This data is also necessary to support your assertion in Line 135. 7) In line 46, recommend revising the phrase "on second place" 8) In the discussion, the two largest case series are discussed and you state that these studies are consistent with "the conclusions drawn from this literature review". This stands to reason as they constitute the bulk of the reported patients. recommend removing or revising this section. 9) In comparing frostbite from nitrous oxide to "standard" frostbite, you site a case series from Texas which, in and of itself, represented an unusual series of frostbite in an center unused to seeing it. Further, within that case series, 7/13 patients underwent surgery. This does not support the statement in your paper, please revise. 10) Line 163 - is there data to support this claim?
|
Thank you for your comments. The authors appreciate your suggestions.
The manuscript has been changed according to comments 1-2.
3. The search did not contain any language restriction, however there was no other article, other than English found.
4. The suggestion 4, has been clarified in the manuscript.
5. There was a wound dressing change at our clinic, one week before it was necrotic.
6. The suggestion 6 has been added to the section limitation.
7. The suggestion 7, it is revised according to the suggestion.
8. The statement has been revised.
9. The statement has been revised. And clarified in the manuscript. The patient should be treated according to the extent of the frostbite. In case of obvious full-thickness frostbite.
10. There are just data from our clinic to support this claim. Although, as we mentioned in the limitation section, unfortunately it was difficult to include patients for a case series due to stigmatization of this particular injury. This is explained in the limitation section. |
The suggestion 9, it is revised according to the suggestion |
Reviewer 2 Report
Comments and Suggestions for Authors
The Authors present a review about an unusual cause of frostbite due to recreational use of nitrous oxide, stating that its frequency has been increasing in the last decade. The paper is interesting and well written and in my opinion has the necessary quality to be published at the European Burn Journal.
There are, anyway, are some remarks that can be made:
A – Major remarks
1. In spite referring, for instance, Chinese references, all the studies included by the Authors in the review came from Europe or USA. It would be interesting to better know the provenience of the patients since this kind of frostbite lesion is not related do geographical or weather features. This information can be I summarized in a table.
2. The same considerations can be made regarding the years of the occurrence of these
particular lesions. In fact, in page 6, lines 132-133, it is assumed an increment of the episodes along the last years but no information is given about the number of patients for each year.
The inclusion of a more comprehensive table or a chart with demographic data referred above could certainly enrich the study.
B – Minor Remarks
Page 2, line 75: “utilizing Rayyan software”: information about intellectual property and copyright (©) must be added to this particular software, as it was made for Excel© in Page 3, line 84.
Author Response
Authors comments Manuscript revision
Reviewer 2: The Authors present a review about an unusual cause of frostbite due to recreational use of nitrous oxide, stating that its frequency has been increasing in the last decade. The paper is interesting and well written and in my opinion has the necessary quality to be published at the European Burn Journal. There are, anyway, are some remarks that can be made:
A – Major remarks 1. In spite referring, for instance, Chinese references, all the studies included by the Authors in the review came from Europe or USA. It would be interesting to better know the provenience of the patients since this kind of frostbite lesion is not related do geographical or weather features. This information can be I summarized in a table.
2. The same considerations can be made regarding the years of the occurrence of these particular lesions. In fact, in page 6, lines 132-133, it is assumed an increment of the episodes along the last years but no information is given about the number of patients for each year.
The inclusion of a more comprehensive table or a chart with demographic data referred above could certainly enrich the study.
B – Minor Remarks Page 2, line 75: “utilizing Rayyan software”: information about intellectual property and copyright (©) must be added to this particular software, as it was made for Excel© in Page 3, line 84.
|
We will correct this in the manuscript. |
All
The copyright information of the Rayyan program has been included in the manuscript. can be found in red text in the manuscript.
We add a sentence regarding the countries from the included studies in the limitation section.
The suggestion regarding patients/year has been included in the result section.
The copyright information of the Rayyan program has been included in the manuscript. |
Reviewer 3 Report
Comments and Suggestions for Authors
The authors have done a review of the literature and reported on a case report of frost bite from nitrous oxide. Whilst they have reported their search terms and inclusion and exclusion criteria, there is no explanation of why they excluded frost bite from medical use of nitrous oxide or indeed why they excluded cold injuries in the mouth as all these are important . If they wish to limit their scope then they need to change the title to include "cutaneous" frost bites.
I would recommend that they revisit their manuscript and use a standardised methodology to write up which should include Boolean operators with the search words; PICO framework and follow the guidelines for reporting literature reviews. They have commented that one of their patients consented to the reporting and they need to include what consent form (institutional?) they have used to obtain written consent from their patient. Their literature review is not comprehensive and it would be advisable to include search items such as "cold burns" and "cold injuries" as well. Furthermore, the value of this manuscript will increase if they were to add other problems with non-medical use of nitrous oxide in their discussion section (such as neuropathies).
Once the manuscript is re-written with the above format in mind it can be reviewed again for consideration of publication.
Author Response
Reviewer 3: The authors have done a review of the literature and reported on a case report of frost bite from nitrous oxide. Whilst they have reported their search terms and inclusion and exclusion criteria, there is no explanation of why they excluded frost bite from medical use of nitrous oxide or indeed why they excluded cold injuries in the mouth as all these are important . If they wish to limit their scope then they need to change the title to include "cutaneous" frost bites. I would recommend that they revisit their manuscript and use a standardised methodology to write up which should include Boolean operators with the search words; PICO framework and follow the guidelines for reporting literature reviews. They have commented that one of their patients consented to the reporting and they need to include what consent form (institutional?) they have used to obtain written consent from their patient. Their literature review is not comprehensive and it would be advisable to include search items such as "cold burns" and "cold injuries" as well. Furthermore, the value of this manuscript will increase if they were to add other problems with non-medical use of nitrous oxide in their discussion section (such as neuropathies). Once the manuscript is re-written with the above format in mind it can be reviewed again for consideration of publication.
|
Author comments: Thank you for your comments. The authors appreciate your suggestions.
In this manuscript the authors wanted to focus on the cutaneous effects of frostbites The authors agree that this should be changed in the manuscript. This manuscript is only a scoping review, although we used a standardised PICO which has now been included in the manuscript. There was also a blinded abstract and full-text screening to have a standardised screening for this scoping review. We include the consent form. We used a written template; we use for all our case report in our institution. This can be sent from the authors if needed. In the section search strategy, we used a search strategy by a professional health science librarian to make this literature review as comprehensive as possible. In the search, there is search items such as “cold burn” and “cold injuries”.
The authors agree with the reviewer, that the value of this manuscript would increase if there would also include non-medical problems such as neuropathies. Since we are plastic surgeons, we wanted to focus on the cutaneous part complications with nitrous oxide. The authors think it is important to focus on this part of complications, and it is therefore limited information of other complications regarding the use of nitrous oxide.
Even though plastic surgery is a technical surgical specialty, we do feel we have certain responsibility to the public, to contribute to preventing injuries from happening, as we do e.g. sunscreen to prevent skin tumours. |
Manuscript revision: We included a PICO in the manuscript. |
Round 2
Reviewer 2 Report
Comments and Suggestions for Authors
The Authors id change the manuscript according to the suggestions and the remarks. Therefore, I have no objections to its publication
Author Response
Thank you for your suggestions!
Reviewer 3 Report
Comments and Suggestions for Authors
The authors have clarified that this is a scoping review of literature rather than a systematic review. I would suggest they remove the word “comprehensive” review of literature as it is now outdated in timeline and no longer comprehensive.
They have added some more references to describe the pathophysiology of cold burns and have included a limitations section. May I suggest that they edit the language of the limitations as there are some minor typos that need correcting although I have ticked the Language box as no edits required!
Overall, much improved presentation and can be accepted for publication.
Author Response
The authors thank you for your comments and suggestions.
Minor language revision has been performed in the Limitation section and we removed the word comprehensive, now calling it only a literature review. This can be found in the revised manuscript with red text.